# Effect of Direct Fluorination on the Transport Properties and Swelling of Polymeric Materials: A Review

**DOI:** 10.3390/membranes11090713

**Published:** 2021-09-16

**Authors:** Nikolay A. Belov, Dmitrii S. Pashkevich, Alexandre Yu Alentiev, Alain Tressaud

**Affiliations:** 1Engineering Center, Tomsk Polytechnic University, 30, Lenin Avenue, 634050 Tomsk, Russia; pashkevich-ds@yandex.ru (D.S.P.); alentiev@ips.ac.ru (A.Y.A.); 2A.V. Topchiev Institute of Petrochemical Synthesis, Russian Academy of Sciences, 29, Leninskii Prospect, 119991 Moscow, Russia; 3Institute of Applied Mathematics and Mechanics, Peter the Great St. Petersburg Polytechnic University, 29, Polytechnicheskaya, 195251 St. Petersburg, Russia; 4Institut de Chimie de la Matière Condensée de Bordeaux ICMCB-CNRS, Université Bordeaux, 87, Av. Dr A. Schweitzer, 33608 Pessac, France; alain.tressaud@icmcb.cnrs.fr

**Keywords:** direct surface fluorination, elemental fluorine, polymeric membranes, gas separation, liquid and vapor permeability, printing equipment, fuel cell membranes

## Abstract

Fluorine-containing polymers occupy a peculiar niche among conventional polymers due to the unique combination of physicochemical properties. Direct surface fluorination of the polymeric materials is one of the approaches for the introduction of fluorine into the chemical structure that allows one to implement advantages of fluorinated polymers in a thin layer. Current review considers the influence of the surface interaction of the polymeric materials and membranes with elemental fluorine on gas, vapor and liquid transport as well as swelling and related phenomena. The increase in direct fluorination duration and concentration of fluorine in the fluorination mixture is shown to result mostly in a reduction of all penetrants permeability to a different extent, whereas selectivity of the selected gas pairs (He-H_2_, H_2_-CH_4_, He-CH_4_, CO_2_-CH_4_, O_2_-N_2_, etc.) increases. Separation parameters for the treated polymeric films approach Robeson’s upper bounds or overcome them. The most promising results were obtained for highly permeable polymer, polytrimethylsilylpropyne (PTMSP). The surface fluorination of rubbers in printing equipment leads to an improved chemical resistance of the materials towards organic solvents, moisturizing solutions and reduce diffusion of plasticizers, photosensitizers and other components of the polymeric blends. The direct fluorination technique can be also considered one of the approaches of fabrication of fuel cell membranes from non-fluorinated polymeric precursors that improves their methanol permeability, proton conductivity and oxidative stability.

## 1. Introduction

The invention of polymerization of tetrafluoroethylene by R. Plunkett’s group in 1938 opened the way for synthesis of semicrystalline perfluorinated and partially fluorinated homo- and copolymers based on the corresponding olefins [1,2,3,4]. The fluorine-containing polymers still retain a separate niche owing to the unique combination of properties (chemical and oxidative resistances, flame retardancy, thermal stability, low permittivity, optical transparency, low adhesion and cohesion, hydro- and oleophobicity) and have found a wide application as materials for chemically resistant components and coatings in chemical processing, pharmaceutical and electrical packaging, biomedical equipment, electrochemical energy storage systems (batteries and fuel cells), etc. [5,6,7,8,9]. They are also used as highly hydrophobic porous membranes for membrane distillation and related processes [10,11].

Partially fluorinated polymers of different classes have been thoroughly investigated as gas separation membranes [12,13,14], although the effects of presence of fluorine in a polymer structure often turned out to be not significant due to its low bulk concentration. Contrarily, amorphous perfluorinated polymers have exhibited outstanding gas transport characteristics, forming upper bonds on Robeson diagrams (i.e., dependence of selectivity on permeability coefficient of the more permeable gas) for He-CH_4_, N_2_-CH_4_, He-H_2_ [15,16,17] (Table 1). However, the synthesis of novel fluorinated and perfluorinated monomers and polymers is complicated by many factors (high cost of chemicals and solvents, difficult and multi-stage preparation of monomers with often low reactivity in polymerization and low molecular masses of the final polymers) that is expressed in the higher costs of some commercial perfluorinated polymers (Teflons AF, Hyflons, Nafion) in comparison with the common industrial polymeric materials (Table 1). This was also observed for fluorine-containing polyimides [18], polymethylenedioxolanes [19], polytricyclononens [20], etc. A convenient way to overcome these restrictions is the enrichment of a thin surface layer of a membrane by fluorine via chemical modification. So, the chemical modification techniques such as chemical functionalization by fluorinated reactants [21], plasma-chemical modification in the presence of volatile fluorine-containing substances (carbon tetrafluoride, perfluoroethane, hexafluoropropylene, etc.) [22], radio-frequency plasma treatments using, in particular, carbon tetrafluoride or octafluorocyclobutane c-C_4_F_8_ [23] and treatment with elemental fluorine [24,25,26,27] can improve operational surface properties of conventional materials.

The previous review on the application of the direct surface fluorination technique focused on adhesion and surface energy, mechanical and electrical properties of the polymers and composites after interaction with elemental fluorine [28]. The current short observation is devoted to a detailed consideration of influence of direct fluorination on membrane properties of the final polymeric materials.

**Table 1 membranes-11-00713-t001:** Physicochemical properties and cost of some commercial fluorinated and non-fluorinated polymers.

Chemical Structure	Abbreviation	Cost ^1^, EUR	*T_g_*, °C	Density, g/cm^3^	He-CH_4_ Selectivity	N_2_-CH_4_ Selectivity	He-H_2_ Selectivity	Ref.
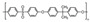	Polysulfone	12	190	1.24	50	0.96	0.93	[29]
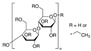	Ethyl cellulose	0.7	110–130	1.14	5.7	0.46	0.71	This work
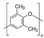	Polyphenyleneoxide	9.5	211	1.06	8.7	0.64	0.62	This work
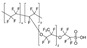	Nafion	486	-	2.0	401	2.5	4.4	[30,31]
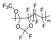 m = 40 mol %, n = 60 mol %	Hyflon AD60	180	125	-	167	3.4	2.9	[32]
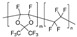 m = 65 mol %, n = 35 mol %	AF1600	384	160	1.78	20	1.3	1.7	[33]
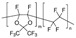 m = 87 mol %, n = 13 mol %	AF2400	400	240	1.67	8	1.4	1.2	[34]

^1^ The costs were extracted from https://www.sigmaaldrich.com/ (accessed on 1 September 2021) and are expressed as prices per gram of polymer.

Transport and swelling phenomena in polymeric materials are important for various applications of membranes, such as gas separation, pervaporation and fuel cells (ion-exchange membranes), barrier polymers for food preservation and electronics and polymers for typographic printing machines and textiles [35,36,37,38]. The mechanism of swelling and transport through a polymer is connected with dissolution of molecules in a polymer matrix with subsequent diffusion depending on the concentration gradient. The rate of penetration of the diffusant is described by permeability coefficient, which is expressed in Barrer (1 Barrer = 10^−10^ cm^3^ cm cm^−2^ s^−1^ (cmHg)^−1^) or in SI units (mol m^−1^ s^−1^ Pa^−1^). The thickness of the selective layer can be often unknown or absent and, in this case, the permeation rate corresponds to permeance in GPU (a Gas Permeation Unit, 1 GPU = 10^−6^ cm^3^ cm^−2^ s^−1^ (cmHg)^−1^). Ideal selectivity, a ratio of permeabilities of gas 1 and gas 2, reflects separation ability of the polymer. Swelling of a polymer in the presence of a vapor is usually estimated by weight uptake or by relative increase in the spatial dimensions of absorbing material. These parameters and their derivatives are related to operational properties of the polymers in aforementioned applications.

## 2. Chemical Composition and Laminate Structure of Fluorinated Membranes

Pioneer works by Nazarov et al. [39] and Koros et al. [40] showed a significant impact of the fluorinated layer on liquid permeation through gas-phase fluorinated polyethylene. Later, the direct fluorination technique was applied for treatment of membranes prepared from different classes of polymers in glassy and rubbery states (polyolefins [39,40,41,42,43,44,45,46,47,48,49,50,51,52,53,54,55,56], polyacetylenes [57,58,59], polyethers [60,61,62,63,64,65], polyesters [66], urethane-based polymers [67], polyetherketones [68], polysulfones [69,70,71], polyimides [72,73,74,75], etc.).

The chemical composition of the fluorinated surface layer depends on a chemical structure of a polymer precursor, reaction time and concentration of fluorine and oxygen (in the case of oxy-fluorination) in fluorinated mixtures with an inert gas [35,76,77]. A common reaction of fluorine with the polymers containing C-H bonds is substitution of hydrogen by fluorine with formation of C-F bonds [78]. During this reaction, highly fluorinated groups can be formed: tertiary and secondary hydrogens are substituted first, while full substitution of methyl group can be achieved after long-term fluorination [56,78]. The C-F containing functional groups can be also generated during fluorination of unsaturated carbon–carbon bonds including aromatic rings [79,80,81]. For the silicon-containing polyacetylenes and polyvinylic polymers, attack of fluorine radical to the silicon–carbon bond leads to (i) homolytic C-Si bond cleavage and subsequent desilylation and (ii) formation of carbon radicals in the main polymeric chain, followed by crosslinking [58,72]. In the case of silicone rubber, Du et al. [82] showed no evident difference in the intensity of IR vibrational bands of Si-CH_3_ (1270 cm^−1^) and Si(CH_3_)_2_ (820 cm^−1^) for short fluorination times (up to 10 min), while prolongation of reaction time to 20 min resulted in a noticeable decrease in Si-C bond content. The authors explained this behavior by a higher dissociation energy of Si-F (584 kJ/mol) and C-F (485 kJ/mol) bonds versus Si-C (347 kJ/mol) bonds. The single carbon–carbon and carbon–oxygen bonds that often take part in formation of polymer backbone seem to be relatively stable at mild fluorination conditions. This fact is proved by a low degradation degree of the fluorinated polymers (see, for instance, Refs. [56,61,80]). The presence of oxygen-containing groups in the fluorinated layers of the polymers is a more complicated case. For a wide range of polymers, the reaction with elemental fluorine leads to the appearance of oxygen-containing functional groups such as C-OH, >C=O, -C(O)F, -COOH even in hydrocarbon-based materials [65,83]. Introduction of oxygen into a chemical structure of polymers can be evidently related to its presence as a contaminant in initial reagents, reaction chamber or to interaction of the fluorinated material with atmospheric oxygen after fluorination [84]. Such interaction might be possible due to the formation of long-lived electron lone pairs in the chemical structure of the fluorinated polymer layer that was detected by electron spin resonance (ESR) measurements [85,86,87]. The carbon–nitrogen bond in the structure of imide and amide groups of various polymers was shown to be rather unstable after interaction with elemental fluorine. For instance, the intensity of C-N bond in C1s and -C(O)-N-H bond in N1s XPS spectra decreased after fluorination of Kevlar fibers, while the intensity of -NF and -NF_2_ peaks in N1s XPS spectra increased [88,89].

A polymeric layer with altered, close to highly fluorinated, chemical structure is finally formed during the fluorination procedure. The depth of fluorination has been shown to be mostly limited, in the case of low permeable polymers, by diffusion of fluorine into the polymeric material. The penetration is evidently higher for highly permeable polymers including rubbers [16]. Nazarov et al. [90] simulated diffusion and simultaneous chemical interaction of fluorine with polymer matrix during the fluorination process and found that the degree of fluorination (the fluorine amount reacted with the polymer in a certain layer relative to polymer surface area) does not depend on the thickness *l*, provided *l* ≥ 4(*D·t*)^1/2^ (where *D* is diffusion coefficient of fluorine in the precursor polymer). Kharitonov et al. have repeatedly demonstrated that the thickness (up to 10 µm) of modified layer is proportional to (*D·t*)^1/2^, and a sharp transient boundary with the original bulk polymer (ca. 0.1 µm) is formed [76,91].

In addition, scanning electron microscopy (SEM) has been utilized to analyze cross-section of the gas separation films from acetyl cellulose (Figure 1) and to estimate the thicknesses of the fluorinated layers which amount to several microns [65]. So, the final product of the direct fluorination has a structure close to a laminate or a composite film.

## 3. Gas Separation Properties of Surface Fluorinated Membranes

### 3.1. Influence of Fluorination Regimes on Gas Permeability

The treatment of polymeric materials with elemental fluorine has been usually performed in various experimental conditions. Researchers have varied the atmosphere of fluorination agent (oxygen, gas oxides), pretreatment (soaking in solvents), other physical impacts (rf-plasma, microwave treatment), temperature, gas or liquid phase regime, etc. However, regarding the fluorination of membrane materials, they have focused on variations of (i) concentration of fluorine in the fluorination mixture with inert gas and (ii) duration of the fluorination procedure.

A common sequence for exhaustive fluorination of organic substances is a progressive increase in fluorine content in fluorination mixture up to 100% and excess pressures [92]. However, in the case of surface fluorination of the polymeric membranes, fluorine concentration in the fluorination mixture has seldom exceeded several percent by volume in order to prevent combustion of the polymer. Systematic evaluation of the influence of fluorine concentration on gas transport properties is summarized in Figure 2. Firstly, it is worth noting the high scatter of the experimental permeance data. This scatter depends on the difference of composite membranes due to randomness of the fluorination procedure and subsequent impurity of the generated fluorinated layer [69]. Even a low concentration of fluorine leads mostly to a sharp decrease in gas permeances of hydrogen and methane for polysulfone (PSF) and poly(2,6-dimethylphenylene oxide-1,4) (PPO) (Figure 2a). However, this trend is not obvious for poly(vinyltrimethylsilane) (PVTMS), a polymer with higher gas permeability (*P*(O_2_) = 40 Barrer) in comparison with PPO (*P*(O_2_) = 15 Barrer) and PSF (*P*(O_2_) = 1.4 Barrer). An increase in the concentration of fluorine results in a gradual decrease in hydrogen permeance while the flux of methane is almost constant (Figure 2a,b). The same behavior is observed for PVTMS, taking into account the scatter of the experimental data (Figure 2a,b). Syrtsova et al. demonstrated that similar methane permeance (*P/l* = 0.05–0.06 GPU) for composite hollow fiber membranes can be attained during fluorination with the fluorination mixture containing 5 and 10 vol.% of fluorine [73]. Langsam et al. showed that total quantity of fluorine passed through the fluorination reactor (involving concentration and flow rate) has higher impact on gas transport than the concentration of fluorine does [58]. This observation can be reasonable if one takes into account that the depth of fluorination depends on the rate of fluorine diffusion in the fluorinated layer. So, low content of fluorine in the fluorination mixture slightly improves O_2_-N_2_ and H_2_-CH_4_ ideal selectivities (Figure 2c,d), while further increase in fluorine concentration either leads to a slight decline in the selectivity or does not significantly change it, leaving the selectivity within the scatter of the experimental data. This decrease is often associated with appearance of defects on the surface of the polymeric membranes during the fluorination procedure. However, the surface defects do not propagate through the entire thickness of the fluorinated layer [58]; this is proved by lower values of gas permeability for the treated membranes in comparison with the virgin ones (Figure 2a,b).

Fluorination duration is the simplest parameter to adjust. The time of the treatment varies from several minutes to several hours for composite membranes with a thin selective layer and up to 100 hours for thick polymeric films [44,54,93]. The formation of a denser fluorine-containing layer should result in a regular decrease in gas permeability. Indeed, a two- to a threefold decrease in hydrogen permeability coefficients are observed for PTMSP and acetyl cellulose (AC), while in the case of PMP and PSF, the *P*(H_2_) declines negligibly (Figure 3a). The more significant effect of fluorination is observed for methane. Its gas permeability decreases in a stepwise manner after the short treatment and proceeds to decrease as the time of treatment increases. The effect of fluorination accounts for the reduction in methane permeability by several orders for LDPE, AC and PTMSP (Figure 3b). For other gases (N_2_, O_2_, CO_2_), the decrease in gas permeability coefficients after fluorination is moderate.

Gas separation characteristics of treated films are also changed in a stepwise manner and either slightly increase or are constant as fluorination time grows. So, O_2_-N_2_ ideal selectivity jumps upward from 1.5 (virgin PTMSP) and is stabilized in the range of 4–5 for the fluorinated PTMSP films (Figure 3c). Contrarily, O_2_-N_2_ selectivity for PMP and PSF slightly decreases as fluorination time increases. For a gas pair with strongly different molecule sizes (for example, H_2_-CH_4_), the effect of direct fluorination seems to be more significant. The ratio *P*(H_2_)/*P*(CH_4_) sharply grows from 1.0 (virgin PTMSP) to 30–100 for the fluorinated samples (Figure 3d). A less dramatic increase in H_2_-CH_4_ selectivity is detected for AC (from 90 to 200–900), PMP (from 9 to 20–30). Interestingly, almost no effect of fluorination is observed for PSF, even for the film treated for approximately 100 hours (Figure 3d).

The duration of surface fluorination should influence the gas separation properties of composite membranes more significantly due to much lower thicknesses of selective layers (up to several microns or less). In fact, a comparison of changes of permeances for PMP and PSF composite membranes with those for the thick films shows the decrease in permeability, in the case of the composite membranes, by a factor of 2–3 (for hydrogen) and 10–100 (for methane) (Figure 4a,b), while a negligible reduction in the corresponding effective permeability coefficients is presented in Figure 3a,b. Similar effects of the direct fluorination duration on permeances are also demonstrated by other composite membranes with a selective layer from PTMSP, PPO, PVTMS and Matrimid (Figure 4a,b).

Fluorination of thin polymeric layers results in peculiar behavior of ideal selectivities. So, a stepwise extraordinary increase in O_2_-N_2_ selectivity is observed for PTMSP, a slight increase in O_2_-N_2_ selectivity with treatment duration is seen for PSF, PMP and PVTMS, while for PPO, O_2_-N_2_ selectivity drops from 4.5 to 2. The latter observation may be associated with a higher defectiveness of the fluorinated composite membranes. The larger scatter of experimental selectivities also indicates this fact. However, these defects do not significantly affect selectivities of gas pairs with small and large molecules, such as H_2_-CH_4_. Namely, H_2_-CH_4_ ideal selectivity is improved during fluorination by 1000 times for PTMSP, and up to 10 times for the other considered polymers (Figure 4d).

Langsam et al. [58] and Le Roux et al. [60,93,94] indicated that the transport properties of composite membranes treated with elemental fluorine are often equal to the properties of a membrane with highly fluorinated thin selective layer. Such behavior can be seen for PSF, PTMSP and PPO in Figure 4. However, the observation is not reproducible for each gas and gas pair.

Direct fluorination of polymers is associated with the generation of long-lived radicals during several months after the fluorination that can react with atmospheric oxygen and then initiate further chemical transformations [87,95]. Therefore, the polymeric membranes may change separation characteristics with time due to this chemical and standard aging processes [96,97]. Surprisingly, no significant alteration of gas permeance and selectivity was detected for surface fluorinated PTMSP-based composite membranes by Langsam et al. [59], although other investigations showed strong aging and reduction in gas permeances for thin PTMSP films [96,98]. Contrarily, a noticeable improvement of He-CH_4_ selectivity during aging was detected for membrane module with Matrimid-based hollow fibers after fluorination. The selectivity increased from 140 (virgin module), through 385 (two days after fluorination), 1090 (9 months) and to 4460 (10 years), while gas permeance decreased from 83 to 37 GPU [73]. The latter selectivity seems to be extremely high for practical application of such surface fluorinated Matrimid-based modules.

### 3.2. Gas Separation Properties on Robeson Diagrams

As it has been mentioned above, the chemical composition of the fluorinated layer is close to perfluorinated polymers and the properties of the layer should be evidently comparable with those for the highly fluorinated or perfluorinated materials. It should be pointed out that gas and vapor transport has been recently summarized for semi-fluorinated polymers by Yampolskii et al. [12,13] and for the perfluorinated polymers by Okamoto et al. [15] and Yampolskii et al. [17]. Besides the aforementioned properties, the presence of fluorine in the macromolecular structure of polymers improves their separation characteristics for specific gas mixtures of interest for natural and biogas sweetening (CO_2_-CH_4_, N_2_-CH_4_), hydrogen enrichment from mixtures of steam methane reforming and methane pyrolysis (H_2_-CH_4_), helium concentration (He-CH_4_, He-H_2_) in combination with lower swelling in the organic solvents [12,13,15]. The perfluorinated polymers are approaching the upper limits for most of these gas pairs on Robeson diagrams [16].

Since the surface fluorinated polymeric film can be considered as a laminate or composite membrane, then the mass transfer through them can be described by models of penetration through the laminates [99] or the composite films [100]. The hypothetic highly fluorinated polymer *B*, representing the polymer which forms the fluorinated layer *1* on top of layer *2* of polymer *A*1–*A*5 (Figure 5a), is assumed to be located on the upper bound for the specific gas pair (Figure 5b). Analysis of selectivity for gas1 and gas2 vs. gas1 permeability (Figure 5b) calculated via resistance model [99,100] suggests that the most promising separation parameters (a combination of permeability and selectivity) for the membranes treated by elemental fluorine are achieved for highly permeable polymers (for instance, *A*1 and *A*2). The plots for the composite membranes exceed the upper limit on the diagram at low fluorination degrees (i.e., low depths of fluorination, corresponding to 0.1–1% of thickness), while the effect of fluorination becomes less pronounced when the permeability of the polymer decreases in a series *A*3-*A*4-*A*5 (Figure 5b).

The predicted behavior is clearly expressed by Robeson diagrams with separation data for the membranes treated by elemental fluorine (Figure 6). Namely, a gradual increase in selectivity toward Robeson’s upper limits for carbon dioxide–methane, nitrogen–methane and oxygen–nitrogen with a slight decrease in gas permeability is observed for virgin and fluorinated ethylcellulose (EC) and polyarylate films (Figure 6a,c,d). The same size-sieving effect is detected for other polymers (polystyrene (PS), copolymer of styrene and acrylonitrile (Co-S-AN), and ABS rubber) [101]. It is worth mentioning that permeability coefficients of gases, presented in Figure 6, are effective and reduced to total thickness of the polymeric film including fluorinated layer. In general, the fraction of fluorinated layer varies in the range of 0.1 to several percent.

Ambiguous results have been achieved in the case of direct fluorination of low-density polyethylene (LDPE) by Kiplinger et al. [44]. A sharp increase in ideal selectivity for helium–methane was obtained after fluorination of LDPE films. The magnitude of the He-CH_4_ selectivity was comparable with that for polyvinyl fluoride (PVF), polyvinylidene fluoride (PVDF) and polytetrafluoroethylene (PTFE) (Figure 6b). The authors assumed that structures of the F-containing ethylene monomers (-CH_2_-CHF-, -CH_2_-CF_2_-, -CF_2_-CF_2_-), including -CHF-CHF- (still not synthesized), occurred during the direct fluorination of LDPE [44]. A less obvious behavior was observed for carbon dioxide–methane selectivity: LDPE films treated with fluorine were less permeable and selective in comparison with PVF, PVDF, PTFE and even with the virgin LDPE (Figure 6a). In contrast to moderate and low permeable polymers, fluorination of highly permeable poly(trimethylsilyl propyne) (PTMSP) resulted in a dramatic increase in ideal selectivity for gas pairs: oxygen–nitrogen (Figure 6d), helium–hydrogen and hydrogen–methane [58]. The data of selectivity permeability for fluorinated PTMSP are close to the actual upper limit or even exceed it in accordance with the predicted behavior (diagram presented in Figure 6b). Detailed investigation of the fluorinated layer of PTMSP by XPS and IR spectroscopy allowed authors to conclude that the fractions of CF_2_, CF_3_ groups and the ratio of F/C increase while the ratio of CH and CF bonds decreases with prolongation of fluorination [58]. The final product of PTMSP fluorination was assumed to have -CF_2_-CF_2_- structure [58] with subsequent removal of -Si(CH_3_)_3_ group as has been mentioned above.

The perfluorinated polymers are located in the middle part of data cloud on the O_2_-N_2_ Robeson diagram, far below the first (1991st) and subsequent upper limits (Figure 6d). Nevertheless, ideal separation characteristics on an oxygen–nitrogen pair for fluorinated PTMSP films, measured in 1988 by Langsam et al. [58], approached Robeson’s upper limit of 2015. A less dramatic increase is observed after direct fluorination of polymeric membranes in gas phase by Chiao [101] (Figure 6d). Such a sharp increase in selectivity is usually associated with a strong size-sieving effect (i.e., with increase in difference between diffusion coefficients of oxygen and nitrogen), typical for low permeable polymers, that may result from a sharp decrease in free volume fraction for the treated polymers [106].

Selective polymers are often used as a thin coating over a porous support (in the form of flat sheet or hollow fiber). Such membranes with selective layers from PTMSP [58], PPO [60], polysulfone (PSF) [69,71,93] and polyimide Matrimid [72,73] were fabricated and fluorinated by elemental fluorine. The separation parameters on CO_2_-CH_4_, He-CH_4_, N_2_-CH_4_ and O_2_-N_2_ for the initial and fluorinated membranes (after fluorination, after surface coating of the fluorinated layer by rubbery PDMS and after aging) were summarized on selectivity vs. permeance diagrams (Figure 7). Most authors have pointed out a decrease in selectivity after fluorination procedure [54,58,60,69,72,73,93]. Composite membranes with PPO selective layer can be a typical example of such behavior: selectivity of CO_2_-CH_4_, N_2_-CH_4_ and O_2_-N_2_ gas pairs for these membranes have usually decreased after the treatment by fluorine, which evidently indicates an increase in defectiveness of the membranes. Healing of membrane surface by silicone rubber significantly increased the selectivity for all presented cases (Figure 7). The data for fluorinated membranes also demonstrate that varying both contact time of the membranes with fluorine and concentration of fluorine in the fluorination mixture has influenced the gas separation characteristics in an equivalent manner [54,56,60] (for instance, see data for PPO membranes in Figure 7).

## 4. Vapor and Liquid Permeation and Related Processes

Trends in gas permeability after fluorination, discussed in the previous section, are observed in vapor and liquid transport through fluorinated polyethylene [39,40,47,49]. Koros et al. have found that the permeability for a wide range of liquid penetrants (alkanes C_5_-C_8_, aromatic hydrocarbons C_6_-C_8_, chloromethanes, 1,2-dichloroethane, dioxane, etc.) through the fluorinated PE layer could be reduced by factors ranging from 1800 to 1.6 × 10^7^, depending on the solubility parameter and the size of penetrant molecule [40]. Permeability and diffusion coefficients of benzene and n-heptane have diminished by up to 2 orders with the increase in fluorination time while the methanol permeability has changed negligibly [39]. Novikova et al. have shown that for n-alcohols (C_1_-C_6_) series, the effect of PE fluorination improved their permeability (except the methanol permeability) as the time of fluorination is increased. The reduction in permeability for other alcohols varied in the range of approximately 50–100 times while that for n-butanol permeability reached approximately 200 [49]. In addition, Manin et al. revealed asymmetry of transport: h-heptane permeability from the side of fluorinated PE layer is twice as low as that from the unfluorinated side (1.0 × 10^−9^ versus 1.8 × 10^−9^ kg·m/(m^2^·s)) [39]. This fact has been explained by a decrease in concentration of n-heptane from the fluorinated side in comparison with its concentration from the side of the virgin polymer. Later on, Wang et al. re-measured permeation of n-hexane, toluene and chloroform and showed a significant decrease in their permeation rate after fluorination by a factor of approximately 14 while the permeation rate of water, acetone and ethanol did not show any apparent difference with the changing F/C ratio for the fluorinated PE [47].

A reduction in permeability coefficients because of the surface fluorination of polymers implies significant improvement of polymer barrier properties. This effect is utilized for modification of coatings for packaging materials (reduction of oxygen and water permeances) [45,47,50,51,87,107,108,109], oil and gas storage and transportation (reduction of fuel permeability) [48,110]. Elkin et al. have suggested a simple treatment of packaging materials (LDPE [109], biaxially oriented PP [108], PETP [107]) by mixture consisting of elemental fluorine (10–20 vol.%), oxygen (10–30 vol.%) balanced by nitrogen, after which the oxygen permeability for the materials is reduced by 20–100%: e.g., from 14 × 10^3^ (pristine LDPE) to 7 × 10^3^ (LDPE film fluorinated for 10 min), from 700 (pristine PP) to 581 (PP film fluorinated for 60 min) and from 100 (pristine PETP) to 28 cm^3^m^−2^bar^−1^day^−1^ (PETP fluorinated during 10 min). A 40–44% reduction in the oxygen permeability for a three-layered packaging material based on PE has been observed by Zazhivikhina and Kolbina [51]. Similar trends were detected after fluorination of PE and PETP [47,50,87,111]. The effect of a significant decrease in vapor permeability can be used for fabrication of polymeric fuel tanks and hosepipes. So, Nazarov et al. demonstrated that preliminary treatment of materials based on HDPE, ethylene-propylene resin, polyisobutylene, butadiene-styrene and butadiene-nitrile-styrene rubbers by common organic solvents (CCl_4_, n-hexane, propanol-2, ethanol, etc.) and by subsequent gas-phase fluorination results in an additional twofold decrease in fuel (gasoline AI92) permeability [110]. This phenomenon can be explained by an increase in interactions between the macromolecular chains due to the formation of polar -CH_2_-CHF-CHF-, -CH_2_-CHF-CF_2_-, -CH_2_-CHF-CH_2_- and similar chemical structures that should limit the diffusion of nonpolar penetrants (for instance, see [112]). An additional factor that may be responsible for the permeability drop can be lower solubility of hydrocarbon penetrants, compared to their perfluorinated analogues, in highly fluorinated substances [13,17,113,114,115]. Meanwhile, the transport of polar vapors, including water, is slightly changed after the fluorination: the water permeability increases or it is almost independent from the fluorination conditions [87,111]. So, Peyroux et al. treated packaging material from LDPE, using various paths, including the single direct gas-phase fluorination and its combination with the plasma modification, ion bombardment and reactive media (oxygen 15 vol.% + nitrogen 85 vol.%) [111]. In all cases, a 20–35% increase in water permeation rate through the modified PE membranes was observed.

The phenomena of vapor and liquid transport in polymers are important for their application as materials for printing industry [110,116]. The researchers aim to improve chemical resistance of the materials toward organic solvents, dyes, moisturizing solutions, to reduce diffusion of plasticizers, photosensitizers and other components through the polymeric blends in order to maintain mechanical and operational properties of the materials [41,42,43,67,117]. Doronin et al. demonstrated that fluorination of butadiene nitrile-based rubber on the surface of ink roller decreased its swelling in washing solvents by 20%, and migration of components (chemical resistance) reduced by a factor of 3 [117]. For typographic plates based on washable polymers (isoprene rubber or urethane-acrylates), the fluorination process reduced swelling in ethylacetate by 50% and 10%, and ten- and fourfold in propanol-2, respectively. Similar results have been achieved by Kamenskaya et al. for moisturizing ink rollers based on butadiene-nitrile rubber [41,42]. The deceleration of diffusion processes (entering of washing solvent and release of low molecular components of polymeric blend) through the fluorinated layer has allowed them to prolong the operational period of the products of the printing industry.

Note that switchable hydrophilic/hydrophobic fluorinated layer was obtained on anodized aluminum for offset purpose by c-C_4_F_8_ radio frequency plasma treatment [116].

Methanol permeability, proton conductivity, the active metallic species crossover/permeability and selectivity H^+^/active species are some of the key parameters for materials used as proton exchange, electrolyte membranes for fuel cell and secondary battery applications. Perfluorinated membranes based on Nafion (DuPont) or, later, Aquivion (Solvay) are traditional materials that satisfy most of the demands for such applications (thermal and oxidative stability, sufficient proton conductivity and mechanical properties), but they are too expensive for widespread use [118] (Table 1). Several approaches have been utilized to reduce the species crossover of the fluorinated membranes, including polymer blending [119], inorganic–organic hybrid [120] and multi-layer membranes [121]. Research interest is also focused on non-fluorinated polymers (for instance, sulfonated poly(arylene ether ether ketones), poly(aryl ether sulfone), polybenzimidazoles, etc.) and post-processing approaches for improvement of membrane characteristics [122]. Direct fluorination of the membrane surface can sufficiently enhance the properties of membranes. Hu et al. [68] and Chen et al. [64] performed direct surface fluorination of polyaromatic composite membranes based on sulfonated poly(fluorenyl ether ketone) and 3-aminopropyltriethoxylsilane. These authors demonstrated that the introduction of fluorine into the fluorenyl structure leads to a significant enhancement of oxidative stability in Fenton’s reagent (approximately twice: from 88 to 182 min), proton conductivity and VO^2+^ permeability (parameter required for utilization in vanadium redox flow batteries), while water uptake slightly declined. Lee et al. [123] modified random disulfonated poly(arylene ether sulfone)-silica nanocomposite membranes by the direct surface fluorination. The treated membranes showed high dimensional stability, low methanol permeability (7.8 × 10^−8^ cm^3^cm·cm^−2^s^−1^), higher proton conductivity (0.157 Scm^−1^) and significantly improved single-cell performances (about 200%) at a constant voltage of 0.4 V in comparison with those of Nafion 117.

Crosslinked polyvinyl alcohol/poly(styrene sulfonic acid-co-maleic acid) membranes have been fluorinated by Kim et al. [124] to reduce water absorption. The depth of fluorination, estimated by XPS analysis (profiles of atomic % changes versus depth), was lower than 10 nm. However, such a fluorinated layer with -CF_2_-, -CH-CF_2_-, -CHF- groups resulted in a decrease of approximately twofold in methanol permeability and an increase in proton conduction efficiency (the ratio of [proton conductivity]/[water uptake]) from 1.6 to 5.9.

## 5. Conclusions

Summarizing the data discussed in the review, the direct fluorination technique can be considered a very effective tool for improvement of membrane properties of polymeric materials. The surface fluorination results in formation of the laminate-structure membranes where the fluorinated layer possesses lower gas permeability and higher selectivity for the selected gas pairs (H_2_-CH_4_, N_2_-CH_4_, He-CH_4_, He-H_2_, etc.). The methanol and water permeabilities are not significantly changed. The effect of fluorination appears often at short fluorination durations (several minutes) and low concentrations of fluorine in the fluorination mixture with inert gas. The commercial non-fluorinated polymers (polyolefins, polysulfones, celluloses, etc.) were treated by the technique, and the most promising results were obtained for the polymers with higher gas permeability. Therefore, further development of materials with enhanced properties requires progress in novel methods for the fluorination of novel polymers with higher free volume, such as polymers of the PIM family [125,126], polynorbonenes and polytricyclononenes [127] as well as substituted polyacetylenes [128]. More detailed investigations of chemical structure and other characteristics of fluorinated layer should be performed on directly fluorinated conventional membrane polymers such as substituted celluloses, polystyrene, polyimides, polyethersulfones, etc. More attention should be also paid to the enhancement of properties of novel, possibly non-fluorinated polymeric membranes for fuel cell and secondary battery applications using direct fluorination processes.

## Figures and Tables

**Figure 1 membranes-11-00713-f001:**
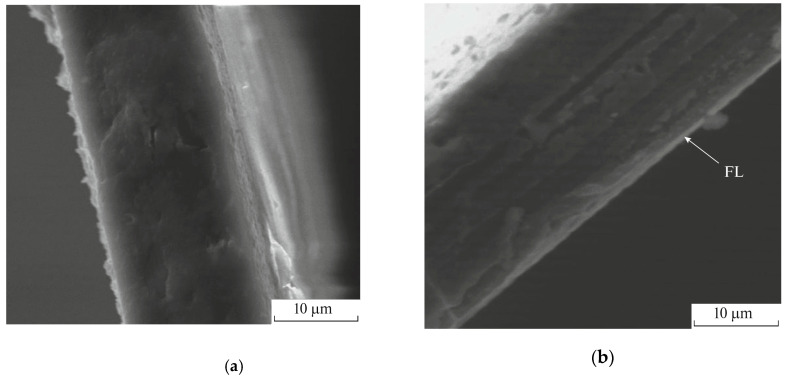
SEM images of (**a**) the virgin acetyl cellulose film and acetyl cellulose films fluorinated for (**b**) 30, (**c**) 60 and (**d**) 120 min. The arrow marks the fluorinated layer (FL). L is the thickness of the fluorinated layer. The figure is adapted from Ref. [65].

**Figure 2 membranes-11-00713-f002:**
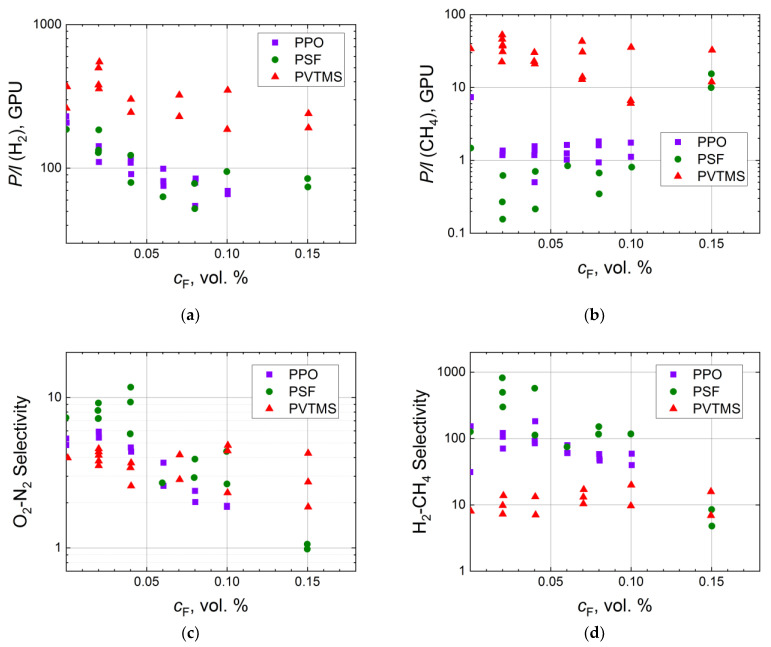
Dependence of H_2_ (**a**) and CH_4_ (**b**) permeances, ideal selectivities of O_2_-N_2_ (**c**) and H_2_-CH_4_ (**d**) for the virgin and fluorinated composite membranes (PPO [60], PSF [93], PVTMS [94]) on the content of fluorine in the fluorination mixture in semilogarithmic coordinates.

**Figure 3 membranes-11-00713-f003:**
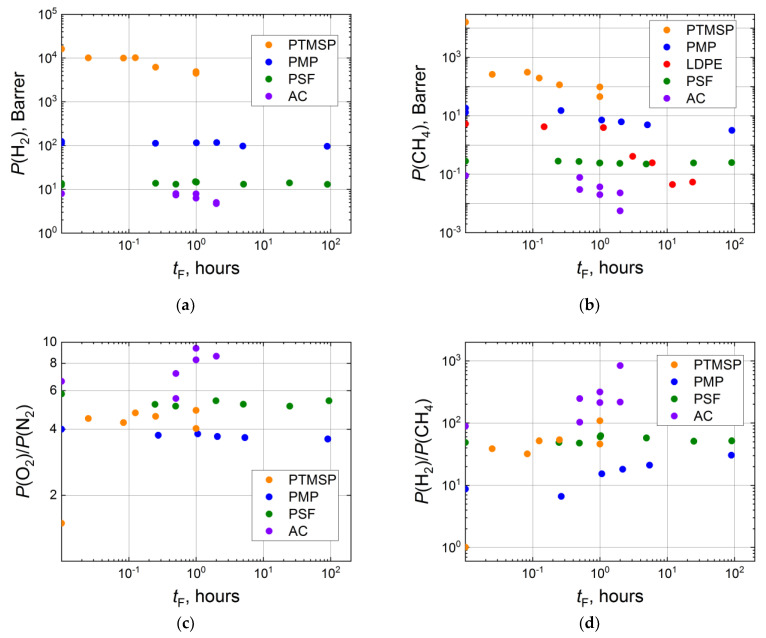
Dependence of H_2_ (**a**) and CH_4_ (**b**) effective permeability coefficients, ideal selectivities of O_2_-N_2_ (**c**) and H_2_-CH_4_ (**d**) for the virgin and fluorinated thick-layer membranes (PTMSP [58], PMP [54], PSF [93], AC [65], LDPE [44]) on duration of the fluorination procedure in logarithmic coordinates.

**Figure 4 membranes-11-00713-f004:**
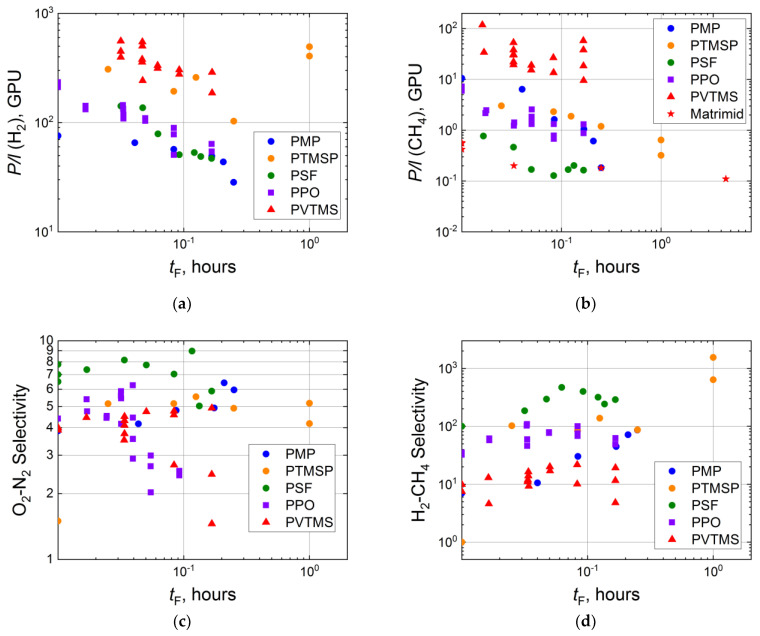
Dependence of H_2_ (**a**) and CH_4_ (**b**) permeances, ideal selectivities of O_2_-N_2_ (**c**) and H_2_-CH_4_ (**d**) for the virgin and fluorinated composite membranes (PMP [54], PTMSP [57], PSF [93], PPO [60], PVTMS [94], Matrimid [73]) on duration of the fluorination procedure in logarithmic coordinates.

**Figure 5 membranes-11-00713-f005:**
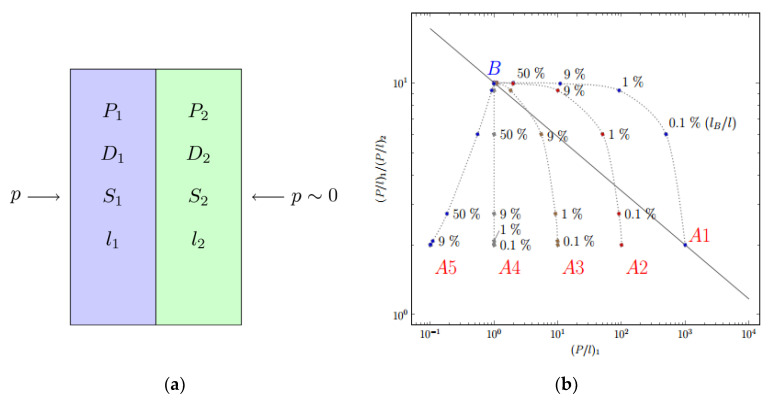
Scheme of two-layer laminate (**a**) and calculated permeability and separation parameters of a composite membrane based on initial polymer *Ai* and hypothetic perfluorinated polymer *B* (**b**).

**Figure 6 membranes-11-00713-f006:**
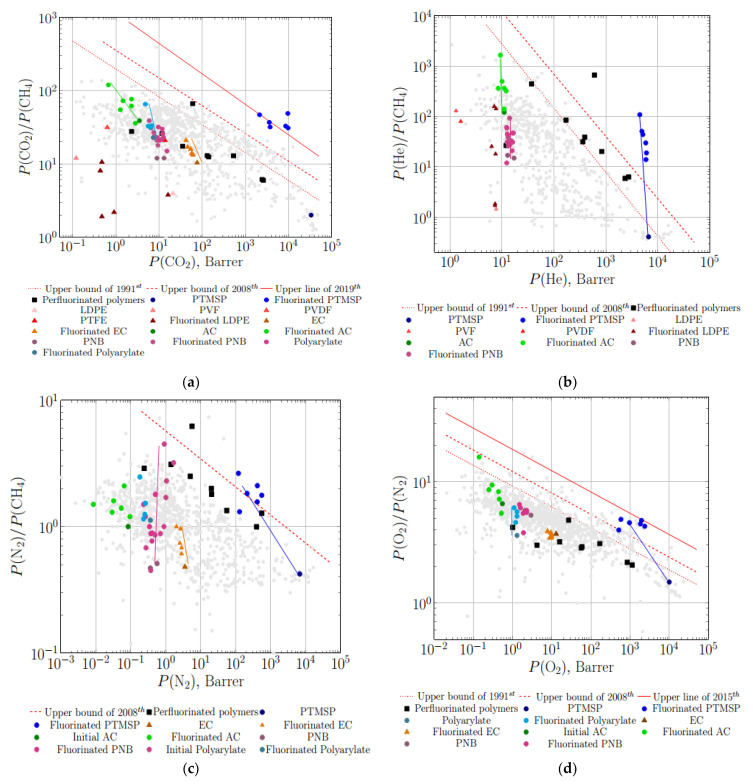
Plots of ideal selectivity vs. permeability coefficient (Robeson diagrams with corresponding upper bounds) for gas pairs CO_2_-CH_4_ (**a**), He-CH_4_ (**b**), N_2_-CH_4_ (**c**) and O_2_-N_2_ (**d**) for original and treated with elemental fluorine films, and data for some perfluorinated polymers: perfluorinated polymers [17,31,102,103,104,105], virgin and fluorinated PTMSP [58], virgin and fluorinated low density polyethylene (LDPE), polyvinyl fluoride (PVF), polyvinylidene fluoride (PVDF) and polytetrafluoroethylene (PTFE) [44], virgin and fluorinated ethylcellulose (EC) and polyarylate [101], acetyl cellulose (AC) [65].

**Figure 7 membranes-11-00713-f007:**
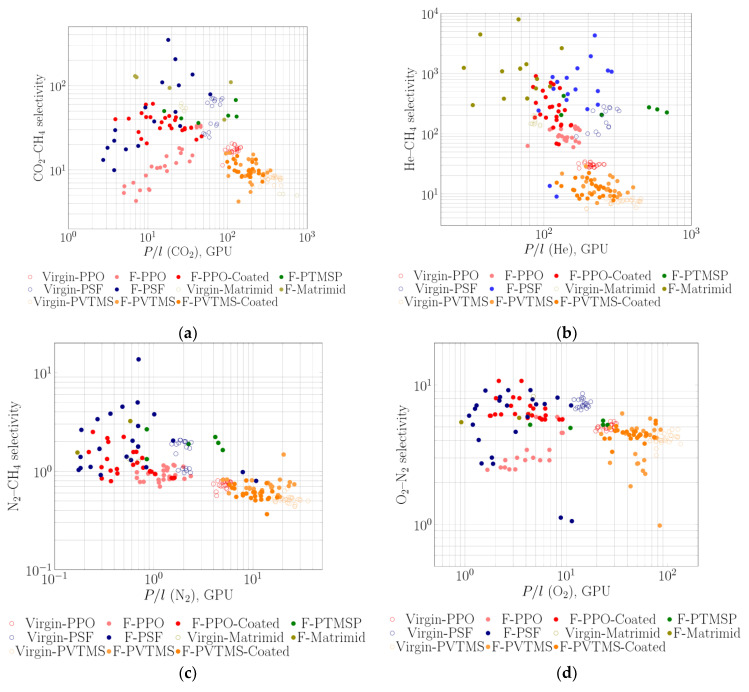
Plots of ideal selectivity of gas pairs CO_2_-CH_4_ (**a**), He-CH_4_ (**b**), N_2_-CH_4_ (**c**) and O_2_-N_2_ (**d**) vs. permeability for the virgin and treated with elemental fluorine and coated with PDMS membranes: PTMSP [58], PPO [60], PSF [69,93], Matrimid [72,73] and PVTMS [94].

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
