# Peer review of "Effect of Direct Fluorination on the Transport Properties and Swelling of Polymeric Materials: A Review"

_membranes, 2021, doi:10.3390/membranes11090713_

Round 1

Reviewer 1 Report

In this manuscript, direct surface fluorination for the introduction of fluorine into the chemical structure of the fluorinated polymers was quite comprehensively summarized and reviewed, which mainly focuses on the influence of the surface interaction of the polymeric membranes with elemental fluorine on gas, vapour and liquid transport as well as swelling and related phenomena. This technique can be also considered one of the approaches of fabrication of fuel cell membranes from non-fluorinated polymeric precursors that improve their methanol permeability, proton conductivity and oxidative stability.

I consider the content of this manuscript will definitely meet the reading interests of the readers of the Membranes journal. However, the discussion is still slightly monotonous, and some comparisons between physical-chemical parameters and cost-performance by accurate values are missing. For the gas selectivity, the discussion is quite enough, while for the vapour and liquid part is relatively short. 

Therefore, I suggest giving a major revision and the authors need to clarify some issues or supply some more data to enrich the content. In particular, the authors should pay special attention to the lack of the definite article the. I will point out some of them, but not all of them.

  1. Abstract and Introduction

  • Line 14, ‘due to the unique combination of physicochemical properties’. Line 16, ‘for the introduction of fluorine into the chemical structure’.

  • Line 20, ‘The increase of direct fluorination duration and concentration of fluorine in the fluorination mixture is shown to result mostly in a reduction of a penetrant permeability, whereas selectivity of gas pairs (He-H2, H2-CH4, He-CH4, CO2-CH4, O2-N2, etc.) increases.’

   This sentence is quite confusing. The reduction of penetrant permeability is toward which species need to be clarified. Does gas selectivity mean one gas can pass, while the other gas in the parish cannot pass, which makes the difference between the two species so that can be called ‘selectivity’ (if so, in the abstract may be ‘gas separation selectivity’ is more understandable)?

  • Line 41, ‘have found a wide application as materials for chemically resistant components and coatings in chemical processing, pharmaceutical and electrical packaging, biomedical equipment, etc. [3–5].’

   Here electrochemical energy storage systems (batteries and fuel cells) should also be mentioned since Nafion is so important in such applications [International Journal of Energy Research 43.14 (2019): 8739-8752; International journal of hydrogen energy 37.7 (2012): 6169-6181].

  • Line 52, ‘This was observed for polyimides [14], polymethylenedioxolanes [15], polytricyclononens [16], etc.’

   This should refer to the above mentioned ‘the synthesis of novel fluorinated and perfluorinated monomers and polymers is complicated by many factors.’ But it seems none of these polymers belongs to fluorinated or perfluorinated polymers. The description should be further clarified to avoid confusion.

  • Line 59, ‘The previous review on the application of the direct surface fluorination technique has focused on...’ Similar problems also exist in other paragraphs and double-check should be made throughout the whole text.

  1. Gas separation properties of surface fluorinated membranes

  • Line 136, ‘An increase in the concentration of fluorine results in a gradual decrease of hydrogen permeance while the flux of methane is almost constant (Figure 2a,b).’

  • Line 156, ‘The formation of a denser fluorine-containing layer should result in a regular decrease in gas permeability. Indeed, a two- to a threefold decrease of hydrogen permeability coefficients are observed...’

  1. Vapor and liquid permeation and related processes

  • Line 293, ‘In addition, Manin et al. have revealed asymmetry of transport: ...has been twice as low as that from the nonfluorinated side... ’

  • Line 338, ‘Methanol permeability and proton conductivity are also some of the key parameters for materials used as proton exchange and electrolyte membranes for fuel cell applications.’

And it should also refer to that, for secondary battery applications, the active metallic species crossover/permeability becomes a critical issue, and the selectivity between [H+]/active species is vital for the battery performance [Journal of Power Sources 493 (2021): 229445]. Several approaches have been adopted to reduce the species crossover of fluorinated membranes, including polymer blending [Journal of Power Sources 196.13 (2011): 5737-5741], inorganic-organic hybrid [Electrochimica Acta 378 (2021): 138133], and multi-layer membranes [ChemSusChem 12.12 (2019): 2620-2627]. The latter referred ‘ VO2+ permeability’ is not a comprehensive summarization since it is only focused on vanadium flow batteries, and it is only one specific method example without mentioning ion selectivity. Indeed, a polymer membrane with only low vanadium crossover is not suitable for battery applications if the proton conductivity is also very low (there will be very high resistance, thus leading to high ohmic loss).

  1. Conclusions

  • Line 362, ‘More attention should be also paid to the enhancement of properties of novel, possibly non-fluorinated polymeric membranes for fuel cell application using direct fluorination processes.’

   It is also suggested to mention and discuss secondary batteries, since not only fuel cells adopt membranes by direct fluorination processes.

  • As a review manuscript, the outlook part should be further enriched.

  • And also about the cost issue, should be further outlooked, since the authors have already referred to the high cost of Nafion in the main text, but what is the cost exactly is not mentioned at all (maybe not in the conclusion part, but in the main text). How about the cost comparison between perfluorinated commercial polymers, partial-fluorinated polymers, and non-fluorinated polymers? Of course, the more F element, the higher the cost, but still some values or numbers need to be demonstrated for comparisons.

   I consider there should also be some comparisons between the physical-chemical properties of perfluorinated, partial-fluorinated, and non-fluorinated polymers, which allows readers to more intuitively understand the difference between chemical properties and performance. I think some tables can be added also for this purpose in the main text part. Now only Figures about gas properties, but not many data shown for vapour and liquid part. Therefore, the overall content of the manuscript is still slightly monotonous.

Author Response

 We are grateful to the reviewer for the thorough examination of the review, useful questions, comments and recommendations. We tried to take them into account and make corresponding corrections and believe that it made the review better.

Comment 1.

I consider the content of this manuscript will definitely meet the reading interests of the readers of the Membranes journal. However, the discussion is still slightly monotonous, and some comparisons between physical-chemical parameters and cost-performance by accurate values are missing.

Answer 1.

Thank you for the comment. The physicochemical properties and cost of several commercial conventional and perfluorinated polymers are included to the manuscript (Introduction, Table 1, highlighted by yellow).

Comment 2.

For the gas selectivity, the discussion is quite enough, while for the vapour and liquid part is relatively short.

Answer 2.

The short discussion of vapour and liquid part (section 4) is associated with relatively low amounts of investigations devoted to liquid permeability for surface fluorinated polymeric materials. This situation is also true for fuel cell and secondary battery membranes.

Therefore, I suggest giving a major revision and the authors need to clarify some issues or supply some more data to enrich the content. In particular, the authors should pay special attention to the lack of the definite article ‘the’. I will point out some of them, but not all of them.

  1. Abstract and Introduction

 Comment 3.

  • Line 14, ‘due to the unique combination of physicochemical properties’. Line 16, ‘forthe introduction of fluorine into the chemical structure’.

Answer 3.

Corrected and highlighted by yellow.

 Comment 4.

  • Line 20, ‘The increase of direct fluorination duration and concentration of fluorine in the fluorination mixture is shown to result mostly in a reduction of a penetrant permeability, whereas selectivity of gas pairs (He-H2, H2-CH4, He-CH4, CO2-CH4, O2-N2, etc.) increases.’

   This sentence is quite confusing. The reduction of penetrant permeability is toward which species need to be clarified.

 Answer 4.

“The reduction of penetrant permeability” concerns all penetrants under consideration in each gas pairs due to formation of barrier fluorinated layer during direct surface fluorination. However, the reduction level is different from one gas to another and this is responsible for change of ‘ideal selectivities’ (a ratio of permeability coefficients or permeances of gas1 and gas2).

The sentence was modified as follows

<The increase of direct fluorination duration and concentration of fluorine in the fluorination mixture is shown to result mostly in a reduction of all penetrants permeability to a different extent, whereas selectivity of the selected gas pairs (He-H2, H2-CH4, He-CH4, CO2-CH4, O2-N2, etc.) increases.>

Comment 5.

Does gas selectivity mean one gas can pass, while the other gas in the parish cannot pass, which makes the difference between the two species so that can be called ‘selectivity’ (if so, in the abstract may be ‘gas separation selectivity’ is more understandable)?

Answer 5.

Polymeric membranes are permeable for all gases but in different manner that results in different separation parameters. The separation properties of membrane are expressed by several coefficients: ‘ideal selectivity’ and different separation factors. See for additional information in Matteucci, S., Yampolskii, Y., Freeman, B. D., & Pinnau, I. (2006). Transport of gases and vapors in glassy and rubbery polymers. Materials science of membranes for gas and vapor separation, 1, 1-2.

The sentence < Ideal selectivity, a ratio of permeabilities of gas 1 and gas 2, reflects separation ability of the polymer.> was added to the Introduction.

Comment 6.

  • Line 41, ‘have found a wide application as materials for chemically resistant components and coatings in chemical processing, pharmaceutical and electrical packaging, biomedical equipment, etc. [3–5].’

   Here electrochemical energy storage systems (batteries and fuel cells) should also be mentioned since Nafion is so important in such applications [International Journal of Energy Research 43.14 (2019): 8739-8752; International journal of hydrogen energy 37.7 (2012): 6169-6181].

Answer 6.

Thank you for the suggestion. We extended the sentence adding the references [8,9] as follows:

<The fluorine-containing polymers still retain a separate niche owing to the unique combination of properties (chemical and oxidative resistances, flame retardancy, thermal stability, low permittivity, optical transparency, low adhesion and cohesion, hydro- and oleophobicity) and have found a wide application as materials for chemi-cally resistant components and coatings in chemical processing, pharmaceutical and electrical packaging, biomedical equipment, electrochemical energy storage systems (batteries and fuel cells), etc. [5–9].>

Comment 7.

  • Line 52, ‘This was observed for polyimides [14], polymethylenedioxolanes [15], polytricyclononens [16], etc.’

   This should refer to the above mentioned ‘the synthesis of novel fluorinated and perfluorinated monomers and polymers is complicated by many factors.’ But it seems none of these polymers belongs to fluorinated or perfluorinated polymers. The description should be further clarified to avoid confusion.

Answer 7.

Thank you for the comment. Of course in this case, we mean fluorine-containing representatives of the polymer types. The sentence was changed as follows:

< This was observed for fluorine-containing polyimides [18], polymethylenedioxolanes [19], polytricyclononens [20], etc.>

Comment 8.

  • Line 59, ‘The previous review on the application of thedirect surface fluorination technique has focused on...’ Similar problems also exist in other paragraphs and double-check should be made throughout the whole text.

Answer 8.

Thank you for the comment. We tried to correct the text throughout the manuscript (marked by yellow).

  1. Gas separation properties of surface fluorinated membranes

Comment 9.

  • Line 136, ‘An increase in the concentration of fluorine results in a gradual decrease of hydrogen permeance while theflux of methane is almost constant (Figure 2a,b).’

Answer 9.

Thank you for the comment. Corrected.

Comment 10.

  • Line 156, ‘The formation of denser fluorine-containing layer should result in aregular decrease in gas permeability. Indeed, a two- to a threefold decrease of hydrogen permeability coefficients are observed...’

Answer 10.

Thank you for the comment. Corrected.

  1. Vapor and liquid permeation and related processes

Comment 11.

  • Line 293, ‘In addition, Manin et al. have revealed asymmetry of transport: ...has been twice as low as that from thenonfluorinated.. ’

Answer 11.

Thank you for the comment. Corrected.

Comment 12.

  • Line 338, ‘Methanol permeability and proton conductivity are also some of the key parameters for materials used as proton exchange and electrolyte membranes for fuel cell applications.’

And it should also refer to that, for secondary battery applications, the active metallic species crossover/permeability becomes a critical issue, and the selectivity between [H+]/active species is vital for the battery performance [Journal of Power Sources 493 (2021): 229445]. Several approaches have been adopted to reduce the species crossover of fluorinated membranes, including polymer blending [Journal of Power Sources 196.13 (2011): 5737-5741], inorganic-organic hybrid [Electrochimica Acta 378 (2021): 138133], and multi-layer membranes [ChemSusChem 12.12 (2019): 2620-2627]. The latter referred ‘ VO2+ permeability’ is not a comprehensive summarization since it is only focused on vanadium flow batteries, and it is only one specific method example without mentioning ‘ion selectivity’. Indeed, a polymer membrane with only low vanadium crossover is not suitable for battery applications if the proton conductivity is also very low (there will be very high resistance, thus leading to high ohmic loss).

Answer 12.

Thank you for the comment. We took into account the reasonable suggestion to include the key parameters of secondary battery application and several approaches to improve operational properties of perfluorinated ion-conducting membranes. They were highlighted by yellow. However, the further consideration of the subject seems to be out of the main theme of the review (direct surface fluorination) and is not comparable with a few number of publications under discussion.

< Methanol permeability, proton conductivity, the active metallic species crossover/permeability and selectivity H+/active species are some of the key parameters for materials used as proton exchange, electrolyte membranes for fuel cell and secondary battery applications. Perfluorinated membranes based on Nafion (DuPont) or, later, Aquivion (Solvay) are traditional materials that satisfy most of the demands for such applications (thermal and oxidative stability, sufficient proton conductivity and mechanical properties), but they are too expensive for a widespread use [118] (Table 1). Several approaches have been adopted to reduce the species crossover of the fluorinated membranes, including polymer blending [119], inorganic-organic hybrid [120], and multi-layer membranes [121]. >

  1. Conclusions

Comment 13.

  • Line 362, ‘More attention should be also paid to the enhancement of properties of novel, possibly non-fluorinated polymeric membranes for fuel cell application using direct fluorination processes.’

   It is also suggested to mention and discuss ‘secondary batteries’, since not only fuel cells adopt membranes by direct fluorination processes.

Answer 13.

Unfortunately, a few amount of investigations were devoted to fabrication of polymeric materials for secondary battery as well as fuel cell applications. That is why impossible to extend the discussion in section 4. And, our suggestion in the Conclusion to develop this application of direct fluorination seems to be reasonable.

Comment 14.

  • As a review manuscript, the outlook part should be further enriched.

Answer 14.

The section 5 is rewritten as follows:

< Summarizing the data discussed in the review, the direct fluorination technique can be considered a very effective tool for improvement of membrane properties of polymeric materials. The surface fluorination results in formation of the laminate-structure membranes where the fluorinated layer possesses lower gas permeability and higher selectivity for the selected gas pairs (H2-CH4, N2-CH4, He-CH4, He-H2, etc). The methanol and water permeabilities are not significantly changed. The effect of fluorination appears often at short fluorination durations (several minutes) and low concentrations of fluorine in the fluorination mixture with inert gas. The number of commercial non-fluorinated polymers (polyolefins, polysulfones, celluloses, etc.) were treated by the technique, and the most promising results were obtained for the polymers with higher gas permeability. Therefore, further development of materials with enhanced properties requires progress in novel methods for the fluorination of novel polymers with higher free volume, such as polymers of PIM family [124,125], polynorbonenes and polytricyclononenes [126] as well as substituted polyacetylenes [127]. More detailed investigation of chemical structure and other characteristics of fluorinated layer should be performed on directly fluorinated conventional membrane polymers such as substituted celluloses, polystyrene, polyimides, polyethersulfones, etc. . More attention should be also paid to the enhancement of properties of novel, possibly non-fluorinated polymeric membranes for fuel cell and secondary battery applications using direct fluorination processes.>

Comment 14.

  • And also about the cost issue, should be further outlooked, since the authors have already referred to the high cost of Nafion in the main text, but what is the cost exactly is not mentioned at all (maybe not in the conclusion part, but in the main text). How about the cost comparison between perfluorinated commercial polymers, partial-fluorinated polymers, and non-fluorinated polymers? Of course, the more F element, the higher the cost, but still some values or numbers need to be demonstrated for comparisons.

   I consider there should also be some comparisons between the physical-chemical properties of perfluorinated, partial-fluorinated, and non-fluorinated polymers, which allows readers to more intuitively understand the difference between chemical properties and performance. I think some tables can be added also for this purpose in the main text part. Now only Figures about gas properties, but not many data shown for vapour and liquid part. Therefore, the overall content of the manuscript is still slightly monotonous.

Answer 15.

The comparison of physicochemical properties and cost of the commercial non-fluorinated and fluorinated polymers were summarized in Table 1 (Introduction).

As we mentioned above, the number of articles on vapour and liquid permeation through surface fluorinated polymers is limited. Moreover, the fluxes are expressed in a different manner and are not reduced often to the thickness. These reasons make comparison of the permeation properties in Table-form complicated.

Reviewer 2 Report

The second review on similar topic after one year seems to be unusually soon. However, the previous review [21] partially by the same authors is commented and differences of this one explained.
I do not feel the paper as very signicifant, but I believe that it may be useful for researchers in this field. I would hesitate with a recommendation for a very prestigious and selective jounal, but I believe that the text should be published in Membranes.

In the text, one article of Okazoe is cited. I think that his article in Journal of Fluorine Chemistry "Development of the “PERFECT” direct fluorination method and its industrial applications" https://doi.org/10.1016/j.jfluchem.2014.09.020 may be also included.
Also Direct Fluorination: A “New” Approach to Fluorine Chemistry, March 2007, DOI:10.1002/9780470166277.ch3 In book: Progress in Inorganic Chemistry, Volume 26 (pp.161 - 210).
Although several papers of Dr. Kharitonov are cited, do the author evaluate the not referred paper in Pure and Applied Chemistry "Surface modification of polymers by direct fluorination: A convenient approach to improve commercial properties of polymeric articles", doi:10.1351/PAC-CON-08-06-02, as irrelevant to the review topic?
  B. Ameduri: Fluorinated (Co)Polymers: Synthesis, Properties, and Applications https://doi.org/10.1002/0471440264.pst575 in Encyclopedia on Polymer Science and Technology

However, I aware that it is never possible to include all related papers and to refer about all; it is a choice of authors what they include in such a review.

The article seems to be nearly free of formal mistakes, which I find in many other submissions sent to reviewing, with one exception: they use "sec" for second instead of standardised "s".
I feel myself not to be qualified to evaluate the language correctness, however the language is well understandable for me as a non-native English user.

After acceptance, I suggest to check subscripts position during proofreading.

I appreciate including the nomenclature list (lines 372 and below) and including DOI in references.

Comments on specific lines:

Lines 35-37: I have not available Ref. 1, but I think that the cited Plunkett's chapter may refer to his original then papers, which should be also referred here, as well as more early papers on the topic.

Line 225: I am bad in English, but what is denoted A3, A4, A5 is connected rather with the word "series" than "row" for me. However, the use might be correct and my understanding wrong - please, check only.
Lines 226-7: Think of making the figure 5b) better readable
Line 470: Ref. 44 is specified insufficiently. If it is a book, publisher should be stated, if thesis etc., the "thesis" word and university should be given, etc.
Line 517: I see mistake in the name of 3rd author in ref. 63, caused probably by a mismatch in character sets during conversion. Please, check during the proofreading
Line 577: Reference 88 is specified insufficiently. See the comment on line 470.

I declare that I am not a co-author of any articles recommended to be cited additionally, neither know personally any of them. I apologize if I have not recognized and have recommended any article already included in the reference list.

Author Response

We are grateful to the reviewer for the thorough examination of the review and fruitful questions, comments and recommendations. We tried to take them into account and make corresponding corrections and believe that it made the review better.

Comment 1.

The second review on similar topic after one year seems to be unusually soon. However, the previous review [21] partially by the same authors is commented and differences of this one explained.

Answer 1.

Indeed, our first review on surface fluorination was published app. one year ago but there are significant difference between the current review and previous one. The first review was devoted to consideration of influence of direct fluorination on surface, electrical, mechanical properties of the polymeric materials. While the current review is focused on detailed discussion of permeation, membrane selectivity, swelling and related properties of the polymers.

Comment 2.

I do not feel the paper as very signicifant, but I believe that it may be useful for researchers in this field. I would hesitate with a recommendation for a very prestigious and selective journal, but I believe that the text should be published in Membranes.

In the text, one article of Okazoe is cited. I think that his article in Journal of Fluorine Chemistry "Development of the “PERFECT” direct fluorination method and its industrial applications" https://doi.org/10.1016/j.jfluchem.2014.09.020 may be also included.

Answer 2.

We thank for the suggestion. The references were included as ref. 27.

Comment 3.

Also Direct Fluorination: A “New” Approach to Fluorine Chemistry, March 2007, DOI:q In book: Progress in Inorganic Chemistry, Volume 26 (pp.161 - 210).

Answer 3.

The reference corresponds to original chapter by Lagow and Margrave of 1979th. It was added to the manuscript as ref. 24.

Comment 4.

Although several papers of Dr. Kharitonov are cited, do the author evaluate the not referred paper in Pure and Applied Chemistry "Surface modification of polymers by direct fluorination: A convenient approach to improve commercial properties of polymeric articles", doi:10.1351/PAC-CON-08-06-02, as irrelevant to the review topic?

Answer 4.

We thank for the suggestion. The references were included as 26.

Comment 5.

  1. Ameduri: Fluorinated (Co)Polymers: Synthesis, Properties, and Applications https://doi.org/10.1002/0471440264.pst575 in Encyclopedia on Polymer Science and Technology

Answer 5.

We thank for the suggestion. The reference was included as third.

Comment 6.

The article seems to be nearly free of formal mistakes, which I find in many other submissions sent to reviewing, with one exception: they use "sec" for second instead of standardised "s".

Answer 6.

We thank for the comment. The “sec” was replaced by “s” in section 4 (highlighted by yellow)

Comment 7.

Lines 35-37: I have not available Ref. 1, but I think that the cited Plunkett's chapter may refer to his original then papers, which should be also referred here, as well as more early papers on the topic.

Answer 7.

Thank you for the suggestion. The ref. 1 was replaced by original Plunkett’s patent of 1941st (R. J. Plunkett, U. S. Patent 2 230 654, Feb. 4, 1941.). Additional reference 2 by R.E. Putnam on history of development of thermoplastic fluoropolymers was introduced.

Putnam, R.E. Development of Thermoplastic Fluoropolymers. In High Performance Polymers: Their Origin and Development; Seymour, R.B., Kirshenbaum, G.S., Eds.; Springer Netherlands: Dordrecht, 1986; pp. 279–286 ISBN 978-94-011-7075-8.

Comment 8.

Question Line 225: I am bad in English, but what is denoted A3, A4, A5 is connected rather with the word "series" than "row" for me. However, the use might be correct and my understanding wrong - please, check only.

Answer 8.

Thank you for the comment. We replaced the word “row” on “series” (highlighted by yellow).

Comment 9.

Lines 226-7: Think of making the figure 5b) better readable

Answer 9.

Thank you for the comment. The size of the signatures on the diagram was enhanced. See novel Figure 5b.

Comment 10.

Line 470: Ref. 44 is specified insufficiently. If it is a book, publisher should be stated, if thesis etc., the "thesis" word and university should be given, etc.
Answer 10.

Thank you for the comment. Ref. 57 is a patent by M. Langsam.

Langsam, Michael. "Fluorinated polymeric membranes for gas separation processes." U.S. Patent No. 4,657,564. 14 Apr. 1987. The reference was modified (highlighted by yellow).

Comment 11.

Line 517: I see mistake in the name of 3rd author in ref. 63, caused probably by a mismatch in character sets during conversion. Please, check during the proofreading
Answer 11.

Thank you for the comment. The surname of the co-author was corrected.

Comment 12.

Line 577: Reference 88 is specified insufficiently. See the comment on line 470.

Answer 11.

Thank you for the comment. Ref. 101 is a patent by C. Chiao.

Chiao, Cherry C. "Surface modified gas separation membranes." U.S. Patent No. 4,828,585. 9 May 1989. The reference was modified (highlighted by yellow).

Round 2

Reviewer 1 Report

I have carefully read the author's revised version and the peer-to-peer replies to the reviewers. I consider all my suggestions for revision have been reasonably replied to and properly solved.

Therefore, I think the current version is acceptable, and I do not have any other revision suggestions.